# Titanium Dioxide: Structure, Impact, and Toxicity

**DOI:** 10.3390/ijerph19095681

**Published:** 2022-05-06

**Authors:** Anca Diana Racovita

**Affiliations:** Department of Chemistry, Faculty of Science, University of Warwick, Coventry CV4 7AL, UK; diana.racovita@warwick.ac.uk

**Keywords:** titanium dioxide, nanotoxicity, industrial applications, photocatalysis, food preservative, sunscreen agent, health controversies

## Abstract

Titanium dioxide, first manufactured a century ago, is significant in industry due to its chemical inertness, low cost, and availability. The white mineral has a wide range of applications in photocatalysis, in the pharmaceutical industry, and in food processing sectors. Its practical uses stem from its dual feature to act as both a semiconductor and light scatterer. Optical performance is therefore of relevance in understanding how titanium dioxide impacts these industries. Recent breakthroughs are summarised herein, focusing on whether restructuring the surface properties of titanium dioxide either enhances or inhibits its reactivity, depending on the required application. Its recent exposure as a potential carcinogen to humans has been linked to controversies around titanium dioxide’s toxicity; this is discussed by illustrating discrepancies between experimental protocols of toxicity assays and their results. In all, it is important to review the latest achievements in fast-growing industries where titanium dioxide prevails, while keeping in mind insights into its disputed toxicity.

## 1. Introduction

Titanium dioxide (TiO_2_) is a white powder extensively used to decontaminate water and food, ensuring environmental and industrial safety, while also serving to protect the skin against harmful radiation [1,2,3,4,5]. To better understand how this metal oxide functions, it is relevant to describe its polymorphic crystal structure [1,2].

### 1.1. Polymorphism

Titanium dioxide exists in three phases: as rutile [1], anatase [1], and brookite [2]. These crystal phases assemble as octahedra, where six oxygen anions are shared by three titanium (IV) cations [2], hence the formula TiO_6/3_, which equals TiO_2_. While expanding in a three-dimensional space, these octahedral units arrange and distort differently for each polymorph, which leads to distinct patterns of crystallinity [2]. As such, the three polymorphs differ in shape, structure [1,2,3], density [1], and refractive index [1]. Rutile has a comparatively higher structural stability [1,2,4,5,6], given that transitions of this phase during synthesis and use are rare [1]. The metastable anatase and brookite can be thermally restructured into the more thermodynamically stable rutile, depending on the mineral’s industrial purpose [4,5]. Brookite is a rarely encountered crystal phase and challenging to manufacture in industry [2].

### 1.2. Optical Properties

Titania (Figure 1) is a white powder with a high refractive index that averages values of 2.7 in rutile and 2.5 in anatase [1,6]. The refractive index is a measure of the metal oxide’s ability to scatter photons [1]. In this sense, rutile reflects light more efficiently compared to anatase and brookite, which marks rutile as an optimal pigment [1]. Interestingly, the metal oxide has a dual behaviour towards light, by scattering and absorbing ultraviolet radiation, both highly desired properties in photocatalysis [1,2,3,7].

Under ultraviolet (UV) light, the metal oxide is a powerful semiconductor, capable of catalysing redox reactions due to its photoelectric properties [1,2,6,8,9,10]. These relate to titania’s band structure as depicted in Figure 2. The energy gap (E_G_) is the region between the valence band, where the lower energy levels are occupied by valence electrons, and the conduction band, where higher energy levels can be filled by more electrons [5,8]. Within a band gap, there are no energy levels for accommodating electrons [2,5]. The value of E_G_ is either equal to or smaller than the energy of ultraviolet photons [5,8]. The region can be associated with a barrier for electron transfer—for higher E_G_ values, stronger UV light must be absorbed by electrons in the valence band to ensure their excitation to energy levels within the conduction band [2,5]. Specifically, titanium dioxide displays E_G_ values of 3.1 eV for rutile and 3.3 eV for anatase [11]. By illuminating TiO_2_ with ultraviolet light, electrons transit from their ground state in the valence band to an excited state in the conduction band (e^−^_CB_), giving rise to positive holes in the valence band (h^+^_VB_) [2,5,8,9]. After this event, molecules adsorbed onto the catalyst’s surface, such as water and oxygen, can undergo redox reactions [2,5,8,9]. Oxidation occurs at the valence band, through h^+^_VB_ electron acceptors, whereas reduction takes place at the conduction band, with the aid of elevated photoelectrons, e^−^_CB_ [2,5,8,9].

### 1.3. Nanoarchitecture Achievements

Titanium dioxide nanoparticles are part of the top five nanoparticles used in industry [12], owing to their versatility in applications—as photocatalysts [5], in pharmaceuticals [13,14], processed foods [15,16,17,18] and household products [13,17,19], cosmetic white pigments [17,18], fabrics [18], and paints, and sunscreens [19].

Compared to microparticles, titanium dioxide nanoparticles display enhanced catalytic activity [3,5]. This is because a decrease in size leads to an increased surface area available for catalysis [5,10,12,20,21]. Recent breakthroughs have been achieved in medicinal applications of photocatalysis, by testing nano-titania as an anticancer agent [14,20,21]. Balachandran et al. reported that irradiated TiO_2_ particles below 20 nm are an efficient “photo-killer” of pulmonary cancer cells [20]. Valence band holes, with their strong oxidant character, lead to the formation of reactive oxygen species; these will interact with defective cells, causing significant intracellular damage, to finally induce their necrosis [20,21].

Modern breakthroughs are also seen in nanobiotechnology. Although many synthetic routes have been designed for nano-TiO_2_, their cost is significant and often associated with environmental hazards [22,23,24,25]. In contrast, the "green" syntheses of nano-TiO_2_ from plants and seeds extracts have been extensively researched, as they prove to be safer, cost-effective, and less toxic [23,24,25]. In general, these methods require TiO_2_ precursors, such as titanium isopropoxide [24] or titanium trichloride [25], which are centrifuged with natural extracts in aqueous solutions [23,24,25]. Interestingly, nanoparticle formation is accelerated by stabilizing interactions with these natural biocomponents [24]. Lingaraju et al. recently tailored the synthesis of anatase titania nanoparticles from fungal biomass [13]. The publication highlights an improved activity of UV-irradiated nano-TiO_2_ against the proliferation of microbial pathogens [13]. Moreover, the metal oxide’s cytotoxic character was assessed, by monitoring the induction of apoptosis in lung and breast cancer cells [13]. Another medicinal attribute observed is the oxide’s role as an anticoagulant, by limiting the formation of blood clots and preventing heart and brain damage [13]. While the interaction mechanisms with cells have yet to be explored, the novel synthesis was facile, cost-effective, and environmentally benign [13].

Titania’s nanoform is also valuable in preventing skin cancer caused by overexposure to ultraviolet radiation [26,27]. TiO_2_ nanoparticles scatter UV photons more efficiently than microparticles [26]. This ability enhances the sun protection factor (SPF) of sun creams, a measure of dermal shielding against photodamage [28,29,30]. Moreover, nano-titania’s photoprotective behaviour [28], coupled with its ability to preserve aliments [31], have seen a multitude of applications in the food industry [15,16,31].

Given the complexity of titania’s features, this review highlights the impact of the metal oxide’s optical properties on the environmental safety sector, and on the pharmaceutical and food industries. Further on, the focus will be on growing concerns in the scientific community regarding titanium dioxide’s nanotoxicity. Then, the discrepancies between toxicity assays will be elaborated on.

## 2. Environmental Decontaminant

### 2.1. Chemical and Biological Warfare Agents

There are countless contemporary threats to environmental and human safety. In this regard, chemical and biological warfare agents (CBWA) have been extensively used as weapons of mass destruction in last century’s war acts, but also in modern times [32,33,34]. Clearing the environment from these toxins has improved in the 21st century, owing to the semiconductor properties of metal oxides such as TiO_2_ [32]. Its ability to generate reactive oxygen species (ROS) in aqueous suspensions has been employed in biological decontamination against ubiquitous Gram-negative pathogens, such as *Escherichia coli* [9] and *Vibrio harveyi,* by cleaving the bacterial membrane and harming intracellular components [35]. Moreover, ROS have been researched for their potential to decompose nerve agent simulants, especially organophosphates, which abound in aqueous environments [32,33].

In this field, doping techniques have seen useful applications by enhancing the photoelectrochemical properties of TiO_2_ [2,3,36,37]. As experimented by Shen et al., the technique incorporates metal ion impurities of different electronegativities and sizes into titanium dioxide’s structure [36,37]. These impurities lower the band gap value, E_G_’, *ergo* increasing the catalyst’s reactivity, as depicted in Figure 3 [2,3,36,37]. As a result, suspensions of the metal oxide’s nanoform co-doped with transition metal cations, such as germanium (Ge^4+^) and zirconium (Zr^4+^), display superior efficiency in degrading chemical warfare agents, compared to pure titania [36]. Shen et al. continued their research on a different titania suspension, doped with Zn^2+^ cations instead of Zr^4+^ [37]. The degradation of toxins was feasible at temperatures below 0 °C, which facilitated catalysis under extreme environmental conditions [37].

Another recent design in this field is represented by water-suspended magnesium micromotors coated with Au-doped titanium dioxide. When irradiated with UV light, the dispersed catalyst generates ROS from water molecules and inactivates *Bacillus globigii*, a simulant for the biological warfare agent *Bacillus anthracis* [32]. The same micromotors were found to decompose organophosphate nerve agents into nonhazardous fragments, indicating an advantageous dual use [32]. The catalytic activity against these warfare agents is enhanced by doping TiO_2_ with Au nanoparticles, efficiently separating e^−^_CB_ and h^+^_VB_ to prevent electron–hole recombinations [32].

### 2.2. Large-Scale Water Decontamination

Degrading persistent chemical pollutants into nonhazardous fragments has also been extensively applied in large-scale water decontamination [38,39,40,41,42]. Various mechanisms of action have been suggested for titanium dioxide [9,43,44,45,46]. The consensus is that once the catalyst is illuminated with UV light of higher energy than its E_G_, molecules that adsorbed on the metal oxide interact with h^+^_VB_, or e^−^_CB_ [2,3,5,8,38], generating a steady flux of hydroxyl radicals (HO^●^), as seen in Equations (1)–(4) [3,9,38,42,43,45]. These radicals can break chemical bonds of molecular impurities, which therefore become non-polluting [9,42,43].
TiO_2_ + hν -> h^+^_VB_ + e^−^_CB_
(1)
HO-OH + e^−^_CB_ -> HO^●^ + OH (2)
OH^−^ + h^+^_VB_ -> HO^●^
(3)
H-OH + h^+^_VB_ -> HO^●^ + H^+^
(4)

Water potability remains a growing concern worldwide, due to the detrimental impacts of chemical spills on human health and environmental balance [3,46]. Over 200 publications in the last 4 years relate to discovering novel methods for water decontamination. This could be motivated by the increasing number and complexity of chemical pollutants [3,5], but also by the persistence of synthetic toxins [46] and medicinal drugs [3] that have already been disposed of in water.

Recent innovations in this field involve the catalytic action of titanium dioxide in nanoform, based on its structural versatility. Even though the catalyst is highly efficient under UV light, there is scientific interest to enhance the compound’s reactivity in the visible spectra, by merging it with carbon-based materials [2,3,39,47]. Notably, Silva et al. improved nano-TiO_2_′s morphology with graphene oxides and carbon nanotubes through thermal treatments [39]. The results have revealed an increased surface area available for catalysis and a reduced frequency of recombination between electron–hole pairs [39], which ensures that more redox sites are active [2,39,47]. The novel approach enhances the efficiency of TiO_2_ upon degrading organic pollutants from both aqueous and aerial media, under both UV and visible light [39].

Leshuk et al. outlined a more facile preparation method for restructured titania, termed “floating photocatalyst", whereby TiO_2_ nanoparticles are mobilised onto a glass microsphere, as seen in Figure 4 [41]. The hollow beads float at random at the air–water interface because they exhibit a lower density than water [41]. Additionally, they can be reused for multiple catalytic cycles, compared to the conventional use of a catalyst slurry [3,41]. This facilitates a cost-effective decontamination, as only a small number of glass beads is required for each water treatment [3,41].

Both syntheses provide structural designs of titanium dioxide with prolonged activity, small costs, and recyclability features [39,41]. This renders the catalyst more efficient than commercially available decontaminants at degrading organic toxins such as benzene [39], dyes [39], and organic acids [41]. One factor that cannot be controlled is the intensity of sunlight in polluted environments.

## 3. Sunscreen Efficacy of TiO_2_

### 3.1. UV Radiation: Exposure Knowledge and Impact

Sun tanning is popular in modern times, not only for cosmetic purposes, but also for medicinal ones [48]. The exposure of skin cells to ultraviolet radiation is beneficial due to the production of vitamin D3 in the body [48] via a complex biosynthesis illustrated in Figure 5. Numerous publications relate vitamin D [49] and its synthetic analogues [50,51,52] to the inhibition of different types of cancer, such as hepatocellular [51], prostate cancer [52,53], and leukaemia [53]. Vitamin D is also responsible for normal levels of calcium [51] and phosphorus [54] in the body, thus maintaining bone strength [48,54] and preventing osteoporosis, a cause of bone fractures [54]. In contrast to the health benefits of tanning, there is a discrepancy between societal awareness of sunburn hazards and preventing them [55]. It has been reported that aesthetic trends and herd mentality take precedence over peoples’ risk assessments of photodamage [55].

Sunburns thus remain a known carcinogen for skin cells [29,55,56]. In this context, there is a high demand for stable, photoactive chemicals termed sunscreens [57], which shield skin cells from the harmful impacts of ultraviolet radiation [29,55,56,57,58,59].

### 3.2. TiO_2_ in Sun Cream Formulations

In general, sun creams consist of organic UV absorbers, whose action is complemented by inorganic UV scatterers [57,58,59,60,61,62]. When combined, dermal photoprotection is enhanced [57,58]. Their combination is necessary because one UV filter alone cannot shield against both ultraviolet B (UVB) and ultraviolet A (UVA) radiation [57,58]. These sunscreens are solubilised with emollients [61] and mixed with fragrances [62], preservatives [62], and many other stabilising agents. The industrial aim is to design cosmetic formulations that are commercially attractive [30,58], safe, and free from allergens [59].

Titanium dioxide is an inorganic sunscreen, authorised by the United States Food and Drug Administration [57] and commercialised under the name Eusolex^®^2000 [30]. The metal oxide scatters UV photons [1,60], thus operating as a shield for harmful radiation [60,63,64]. However, the dual reactivity of titania towards sunlight is problematic in the sunscreen industry, given that light absorption can activate the catalyst and lead to redox reactions between h^+^_VB_ or e^−^_CB_ [9] and the surrounding elements within the sun cream [60,63,64]. Any structural changes in the formulation matrix are undesirable due to their impact on the cream’s stability, safety, and photoprotective function [60,63,64]. To prevent photocatalysis, an extensive body of literature is dedicated to weakening titanium dioxide’s catalytic performance whilst preserving its light-scattering attribute [60,63,64,65,66,67,68]. Thus far, this has been achieved by coating the nanomaterial’s surface with an unreactive shell, such as aluminium oxide (Al_2_O_3)_ [60], zirconium dioxide (ZrO_2_) [60], silicon dioxide (SiO_2_) [60,63,68], silicon tetrahydride SiH_4_ [63] or polymers of diethoxysiloxane [67].

### 3.3. Silica-Coated, New-Generation TiO_2_

Silicon dioxide represents a useful inert layer, based on its insulating properties [66]. Several methods for coating TiO_2_ with SiO_2_ have been designed. One such method, the solvothermal process [63], is supported by El-Toni et al. The titania nanoparticles are coated with silica via a facile, mild, and time-efficient process that enables control over the shell’s thickness and structure [63]. The cover of silica insulates titanium dioxide’s catalytic surface underneath, therefore inhibiting undesired redox reactions in the presence of UV light [63]. Because the inert coat is transparent, UV light will still reach TiO_2_ [69] and be scattered. The oxide’s role as a sunscreen is maintained, while safety has improved [60,63].

Another methodology implemented for silica coating is the sol–gel technique [68]. Jaroenworaluck et al. utilised the same source of silica and reagents as El-Toni et al., but lower temperatures, slower mixing of chemicals, and longer reaction times [68]. The coated titania exhibited enhanced light-absorbing properties compared to the industrially available, uncoated material [68]. However, the suppression of photoactivity was not established [68]. Furthermore, although both techniques were experimented at the nanoscale [63,68], the coated TiO_2_ resulting from the solvothermal process displayed a homogenous silica layer [63], compared to the coat formed via the sol–gel technique [68]. Thus, El-Toni et al. confirmed that higher temperatures and control over silica content influence shell layer uniformity, which is beneficial in the photoinhibition of nano-TiO_2_ [63]. The field remains promising for future improvements because it is challenging to inactivate titanium dioxide’s catalytic activity completely [60,63,67].

## 4. TiO_2_ Impact in the Food Industry

### 4.1. Smart Packaging Composite

Titania’s high refractive index and reactivity towards UV light [1] are remarkable properties not only in the sunscreen industry, but also in food processing [15,16,31,70,71,72]. The oxide’s nanoform has been developed for use in packaging light-sensitive food products [15], to ensure food durability. In this regard, merging TiO_2_ nanoparticles and biopolymers has given rise to “Active Food Packaging (AFP)” [72,73,74]. AFP displays increased mechanical strength and antimicrobial properties [16,70,71,72,73,74,75] compared to other "green" wrapping materials, such as paper, which is more sensitive to tearing, attracts moisture, and allows microbes to proliferate [70].

The biolayer is usually chitosan [70], or a derivative [16] that has an intrinsic biocidal activity [76]. The biolayer is permeable to water and oxygen, which are adsorbed on the package’s surface, diffusing through it [16,72]. In the presence of UV light, TiO_2_ transforms the diffused molecules into reactive oxygen species [16,70,72,74]. These species target, oxidise, and inactivate microbes that would otherwise proliferate in food products [16,71,72]. ROS were also found to contribute towards the photodegradation of ethylene [16,77], a highly active growth regulator for fruits and vegetables [77,78]. As such, fresh food discolouration [78] and accelerated food softening [77,78] are also minimized. The AFP mechanism of action has been summarized in Figure 6.

One impediment observed experimentally by Siripatrawan and Kaewklin is that, during AFP manufacturing, TiO_2_ nanoparticles cluster with increasing concentrations [16]. The formation of aggregates decreases titania’s catalytic efficiency, due to a decrease in its effective surface area [3,5,16,20,79]. This impediment, also supported experimentally by Li et al., affects the rate of redox processes required for microbial inactivation [16,79]. To choose optimal AFP candidates, multiple synthetic trials are necessary, in order to compare the materials’ catalytic efficiencies [16,79].

Overall, merging TiO_2_ and biomaterials prolongs food storage, while preventing health hazards associated with contaminated food [71,72,73,80]. In addition, chitosan’s biodegradable polysaccharide structure [70] lowers the environmental footprint specific of conventional, artificial packaging [15,72,73].

Another advantage of AFP is that it functions as an antimicrobial without releasing any chemicals onto the food supply [72,73,75,80]. In contrast, direct contact between biocides and consumer goods has led to health concerns. The diffusion of antimicrobials from the food surface into the bulk can alter the product’s taste [73] or structure [80]. Spoilage, regardless of degree, is not accepted by the European Food Safety Authority (EFSA) regulations for food safety [81]. Moreover, reactive chemical release is counterproductive, because food contamination is an issue to be resolved by antimicrobial agents [81], not caused by their action. In comparison, AFP is part of an expanding area of research. Smart packaging is more efficient than previous preventive methods against food contamination because it irreversibly damages fungal and bacterial agents through a localised pathway [80]. Table 1 summarises the bactericidal and antifungal activity of smart composites against a wide range of food contaminants.

### 4.2. Food-Grade TiO_2_

Consumer trends and dietary habits are diverse and changing at a fast pace [91]. These are mainly influenced by modern lifestyle choices and by societal demands for a diversity of new flavours [91]. Consequently, safe preservatives are necessary for tackling the waste of fresh foods and beverages, while enhancing their physical properties [91].

In the food processing sector, titanium dioxide comes under the name E171 in the European Union and INS171 in the United States [92]. The name “titanium white” [6,93] is also significant, because the metal oxide powder increases the opacity, brightness, and whiteness of many foods [93,94,95,96]. As such, it is authorised for use in creamer formulations, [94,95,96,97] to thicken their texture, and in fish [96,97], instant drinks [94], and confectionary products [93,95], to enhance their flavour. It was also found to improve the taste of many coloured foods incorporated into a daily diet [93,96,97]. Despite its presence in many foods, E171 is an additive with no nutritional value [97].

### 4.3. Emerging Controversies on E171

In the late 1960s, the European Union and the United States Food and Drug Administration marked food-grade titanium dioxide as safe [92]. However, in the last decade, there has been a concerning rise in publications on food-grade TiO_2_. Its practical uses in the food sector are often associated with warnings on the health impact of nanosized E171 [93,94,95]. The plethora of publications in this field may have been propelled by the lacunes in European Food Safety Authority’s (EFSA) re-evaluation study of E171 safety, as demanded by the European Commission [98]. The panel’s target was to accurately rate the safety of titanium dioxide in food products, following oral administration [98]. In 2016, the conclusion reached by the committee is that although ingestion of E171 does not lead to carcinogenesis, its nanotoxicity on the reproductive system is inconclusive [98]. At the time, no further steps were taken by EFSA to elucidate the link between E171 and the reproductive toxicity, and an acceptable daily intake of the food additive was not established [98].

Another influencing factor for the high number of publications in this area is the analytical determination of an accurate risk assessment for nano-TiO_2_ [93,99,100]. Currently, there is no documented, legally approved approach for testing particle size distribution (micro- and nano-) in E171 or in consumer goods exposed to E171 [93,94,95]. Concerningly, small-size nanoparticles, showing high reactivity [20,21], were found to elude detection even with modern analytical tools [93]. Ideally, E171 would only contain microscale titania to achieve the desired effects in food processing [99]. However, scientific studies highlight the presence of nanoparticles in E171 in alarming concentrations, which poses a health hazard [93,94,96,99,100]. Already in France, since January 2020, E171 is temporarily banned by decree [99]. In May 2021, relevant scientific proof of titania’s genotoxic behaviour, when accumulated in the body, was taken into consideration by the EFSA committee [100]. The panel re-emphasised the impossibility to quantify an acceptable daily intake of the preservative [100]. However, once E171 has been exposed as a potential genotoxin, a safety threshold for administration cannot be determined, because the chemical should not be administered in the first place [100]. The European Commission is now responsible for re-evaluating the legislation around E171 use [100].

Suffice to say that EFSA’s decision on the topic is the most valuable regarding regulatory changes on nanomaterials in food industries [98,100]. Yet, the French government has already acted on concerns in the scientific community regarding E171 toxicity, by imposing a legal ban on its use a year before EFSA’s assessment [99,100]. The nanotoxicity of E171, in all its complexity, was being investigated five years ago, as it is now. A prompt response on limiting its use is vital for ensuring the population’s safety.

## 5. Titania Toxicity

The downside of nanoreactivity is herein described, by reviewing the methodologies employed in toxicity studies concerned with titanium dioxide. The latter’s damaging potential after absorption and distribution in the body, as well as controversies around the mineral’s toxicity, is the core of this section.

### 5.1. Post-Inhalation Toxicity

Late-20th century publications on titanium dioxide’s toxicity exposed a correlation between inhaling fine particles of this mineral and the emergence of pleural diseases, especially in workers exposed to it in manufacturing plants [101,102]. At present, titanium dioxide is classified by the International Agency for Research on Cancer (IARC) as a 2B-type carcinogen, equivalent to “possibly carcinogenic to humans” [103]. This classification also implies that further studies are required to fully ascertain its cytotoxic and genotoxic potential [103].

According to clinical research, many experiments have been performed on rodents to strengthen the link between the inhalation of TiO_2_ and the development of lung tumours [104], lung inflammation [105], breakdown of alveolar membranes as a result of the body’s immune response [106], as well as lung tissue scarring [107]. However, there are many trials that implement questionable protocols [105,106,108]. Warheit published an extensive review on protocol discrepancies [108]. The main findings support that carcinogenesis in murine lungs is influenced by the metal oxide’s particle size, crystallinity, and overload in the lungs, thus not directly or solely related to its reactivity towards pulmonary cells [108].

In toxicity assays, methodologies of relevance must be able to simulate realistic conditions, instead of introducing extreme parameters that are bound to reveal a detrimental impact. For example, in the protocol designed by Chen et al. [106], nano-TiO_2_ suspensions were instilled into tracheal incisions of rodents until detrimental pulmonary changes were recorded [106]. This differs significantly from human exposure conditions. In a quality-accredited industrial laboratory, tests are performed on the metal oxide powder under rigorous health and safety regulations. These are in place in order to protect manufacturers or analytical scientists, who are trained to abide by testing procedures. To assist this, the largest study on human lung exposure to TiO_2_ concludes that neither the occupational history of manufacturers nor their exposure to TiO_2_ during employment is related to developing lung carcinoma [109].

To date, there are insufficient data on the duration of human lung exposure to nano-TiO_2_ and the ensuing effects to make a relevant scientific correlation [101,102]. Therefore, extrapolating results from rodent studies [104,105,106,107,108], with little control over parameter rigour, does not represent an accurate evaluation of nano-TiO_2_ toxicity in the human respiratory system.

### 5.2. Post-Ingestion Toxicity

For almost a decade, oral ingestion of nano-titania has represented a concerning health issue for the population [110,111]. Studies on rodents and human volunteers indicate that once titania passes the intestinal barrier, it reaches the bloodstream and is then transported to vital organs [97,110,111]. If oral consumption of the oxide’s nanoform exceeds the recommended limit, it can accumulate in tissues and induce irreversible damage [112]. The harmful effects observed on rodents include stomach and intestine inflammation [113], liver cell necrosis [112], lesions to cardiovascular tissue [112], enhancement of anxiety disorder [110], progression of colon tumours [114], and nephrotoxicity in kidneys [112]. There are many experimental variables that can influence the outcome of ingestion toxicity assays, but the main ones to consider are the concentration and reactivity of TiO_2_ nanoparticles.

#### 5.2.1. Concentration Discrepancies

The concentration parameter is crucial to the outcome of toxicity assays and therefore must be chosen carefully for in vivo experiments.

The in vivo study carried out by Wang et al. exposed rodents to a high, fixed dose of 5 g/kg_bw_ of TiO_2_ per day over the course of fourteen days, showing deleterious effects on hepatocytes, nephrons, and on myocardial tissue [112]. However, inconsistencies to real life are notable when comparing the experimental protocol [112] to European regulations [98]. Realistically, the human oral daily intake of titania nanoform results from ingesting E171 and was estimated by EFSA to be between 0.5 and 14.8 mg/kg_bw_ [98]. This estimate is approximately 1000-fold smaller than the quantity employed by Wang et al. in the rodent study. Another parameter influencing the experiment is the method of administration, which registered accidental flaws. Rodents were exposed to titanium dioxide suspensions by being forcefully fed a single daily dose, which led to mortality by mistake in some of the subjects [112]. These undesired variables, though mainly kept under control, make the study weakly reliable for human risk assessments of oral exposure to nano-TiO_2_, given that human digestion is a more complex, gradual, and timely process [115].

In contrast, the in vivo toxicity experiment conducted by Medina-Reyes et al. exposed rodents to 5 mg/kg_bw_ of nano-TiO_2_, which can simulate the human oral intake of this mineral for different age groups [98,110]. The rodents’ health levels were carefully monitored over sixteen weeks instead of two [110,112]. Another important factor considered in this study is the dieting behaviour of the rodents [110]. The latter were organised into high-fat- and regular-fat-diet categories, which enabled the researchers to compare the effects of nano-TiO_2_ ingestion on both categories, by analysing injuries to the gastro-intestinal tract, liver, and reproductive system [110]. The administration pathway also differs from the study carried out by Wang et al. [112]. Instead of forcefully feeding the rodents with a high fixed dose of the metal oxide slurry [112], E171 was introduced in the rats’ drinking water, alongside their diet [110]. This gradual exposure allowed for a realistic simulation of human digestion steps and substantially improved the test results’ reliability.

#### 5.2.2. Human Toxicity Models in Different Concentrations

Novel in vitro techniques in the field of toxicity assessments focus on simulating human exposure to E171 with better accuracy than murine models [116,117]. Notably, Dorier et al. exposed a model of human intestinal cells to food-grade titanium dioxide and pure nanoparticles in an environment relevant to human physiology [116,117]. The studies investigated the link between the nanoparticles’ reactivity, the formation of reactive oxygen species (ROS), and intracellular oxidative stress, which can lesion deoxyribonucleic acid (DNA) strands and induce carcinogenesis [15,116,117]. The results have revealed that, regardless of the nano-TiO_2_ concentration, human intestinal cells are protected from the nanoparticles’ reactivity even when the cells are at their most sensitive phase of development, i.e., before differentiation to meet specific functions in the body [116,117]. E171 thus only displayed moderate genotoxicity due to ROS formation [116,117]. The technique of continuous exposure has shown better accuracy compared to an acute, time-limited exposure [117] to correlate in vitro and in vivo studies [116].

### 5.3. Toxicity after Dermal Exposure

Stratum corneum cells reside in the uppermost layer of the skin, covering the entire human body [12,118]. These cells represent a significant route for absorption of nano-TiO_2_ [12,118], which is incorporated into the photoprotective medical make-up [119] and sun cream formulations [12,58,59,60]. In this context, both in vitro and in vivo studies question the harmfulness of cosmetics containing titanium dioxide [12,120,121,122,123]. The findings are concerning in the scientific community because sunscreens are designed with the aim to protect the skin from carcinoma [58,59] and not be the cause of mutagenesis.

First, controversies have appeared regarding the ability of nanosized titanium dioxide to penetrate both the epidermis and vascularised dermis, when applying sun creams topically [12,28,59,118,119,121,122,123,124]. As indicated previously, when dispersed in the circulatory system [59,110,111], the nanomaterial can accumulate in organ tissues, leading to deleterious side-effects [110,111,112,113,114,116,117].

Secondly, when applied on the skin, UV-illuminated titanium dioxide can easily contact moisture in the atmosphere [97,120]. As a result, e^−^_CB_ and h^+^_VB_ [2,8,9] catalyse the formation of reactive oxygen species from water and oxygen [12,18,28,123]. These include hydroxyl radicals [8,43,44] (Equation (4)) and hydroxyl anions (Equation (3)). As seen before, their formation is significant in environmental and food decontamination [16,32,33,34,35,70,71,74,75]. However, the same ROS were exposed as a mutagen for dermal cells [26], by causing DNA strand breaks and oxidative stress [65,120,123].

Although coating the material with inert shells may delay titanium dioxide’s catalytic activity, an irreversible inhibition of its reactivity is difficult to achieve [63,67,121]. Furthermore, the crystallinity of TiO_2_ nanoparticles in sun creams can also influence the catalytic effect [64,68,121]. While rutile scatters light most efficiently [1], anatase absorbs photons more strongly [64,121]. A better regulation over the polymorphs incorporated into sun cream formulations seems necessary in order to inhibit the metal oxide’s catalytic power.

## 6. Titania Toxicity Controversy

Owing to the great variety of applications of titanium dioxide, especially at the nanoscale, it is necessary to investigate its genotoxic and cytotoxic potential. At present, consistent results between toxicity assays are challenging to achieve [125,126,127,128,129]. Naturally, different experimental parameters and methodologies focus on distinct types of cellular damage [127,128,129,130,131,132,133,134,135,136,137]. Table 2 summarises the most encountered in vitro toxicity assays for titanium dioxide nanoparticles.

### 6.1. Parameters in Need of Improvement

Comparisons between in vitro toxicity assays for nano-TiO_2_ reveal that inconsistencies are more frequent for this catalyst than for other inorganic metal oxides or precious metal nanoparticles with conductive character [127]. The exposure of titanium dioxide to UV light in the laboratory seems to be responsible for experimental inconsistencies, as irradiation can trigger its photocatalytic activity [127,138,139]. Moreover, although an increased surface area from the microscale to the nanoscale is beneficial in some industrial applications [5,12,20], the same role allows the catalyst to interact with reagents or equipment utilised in assays, thus interfering with experimental endpoints [2,117,127,129,130,131,132,133].

### 6.2. The Dichlorofluorescein (DCF) Assay

In the DCF assay, the concentration of intracellular reactive oxygen species formed after cellular exposure to nano-TiO_2_ is determined by the extent of oxidation of precursor H2DCFDA to fluorescent DCF, as depicted in Figure 1 [116,129,131,132,133]. The fluorescence intensity of DCF is then detected via spectrofluorimetric measurements and is proportional to the concentration of reactive oxygen species [12,131]. Photomicrographs help visualise and compare control cells with cells exposed to nano-TiO_2_ [12].

When quantifying the levels of ROS, for example, in alveolar [129], epidermal [12], and hepatic cells [132], the fluorescence intensity of DCF was found to vary, depending on the exposure levels to the white mineral [12,129,132,133]. The study carried out by Kroll et al. reveals that the fluorescence emission intensity decreases in cellular media with an increasing concentration of the metal oxide because of its light-absorbing properties [129]. The finding is also confirmed by Aranda et al. [132]. On the contrary, the experiments carried out by Guadagnini et al. and Shukla et al. argue the opposite outcome, that the fluorescence intensity of DCF increases with an increasing concentration of TiO_2_ nanoparticles [12,133]. In either case, the influence of titanium dioxide on spectrofluorimetric measurements was notable at higher concentrations of the mineral and longer cell exposure times [12,129,132,133]. The experiments prove that the DCF technique is liable to either false-positive or false-negative endpoints, when employed for toxicity assessments of light-sensitive nanoparticles [129,132,133].

### 6.3. The MTT Assay 

When the yellow MTT dye comes into contact with cells, their metabolic activity reduces this salt to the purple MTT-formazan (Figure 2) [133,134,140].

The latter is then quantified via colourimetry, a technique that relates the MTT-formazan absorbance to its concentration with Beer–Lambert’s Law [141,142]. As such, the concentration of purple dye will indicate the population of viable cells whose metabolism was unharmed after exposure to nano-TiO_2_ [117,129,133]. Often, the MTT assay is also prone to inaccuracies in titania toxicity reports [129,133]. For example, Kroll et al. ascertained that at high concentrations of titanium dioxide nanoparticles, the absorbance of the MTT formazan dye increases significantly [129]. In an experiment, this would lead to false-positive results, by detecting a higher cell viability post-exposure to the catalyst [129,133]. The outcome conforms experimentally with the findings of Guadagnini et al. [133].

### 6.4. The Neutral Red (NR) Assay

This method monitors the ability of healthy cells to incorporate neutral red dye into their lysosomes [136]. The supravital stain is extracted and then colourimetry detects the concentration of cells that remained intact post-exposure to the metal oxide [140]. Although this assay measures a different endpoint, it is more cost-effective and sensitive than the MTT assay [136,140]. However, light distortion was observed in the NR assay carried out by Guadagnini et al. [133], owing to titanium dioxide’s dual behaviour towards light [1]. The results expose an increase in the dye’s absorbance in the presence of the metal oxide and absence of cells [133]. This can impact the end point and, during a cytotoxic assay, an erroneous, higher cell viability would be recorded.

### 6.5. Meta-Analysis as a Preventive Action

Recently, novel approaches are seen through meta-analyses of experimental results. Many toxicity assays published so far for nano-TiO_2_ have been analysed in order to obtain a trend in the most influential factors during experiments [125,126]. These include the size range of the nanoparticles [125,126], coating materials [60,63,68], concentration [125,126,129], and crystal phase [143], all of which influence photoreactivity. Ling et al. conducted an up-to-date comprehensive assessment of 1592 publications related to the in vitro nanotoxicity of titania [125]. Filtering was performed by excluding studies that did not offer (i) the aforementioned factors of influence, (ii) duplicate results of assays, in agreement with legislation requirements, and (iii) clear, implemented protocols [125]. As such, screening allowed for the selection of 59 studies of high-quality score [125], which approximates to 3.7% of the total number of publications at the start of the meta-analysis. A similar screening system for toxicity assays on titanium dioxide was applied by Charles et al., which allowed for the selection of 36 relevant studies out of 100 [126]. The numbers suggest that an in-depth classification of relevant experimental conditions is vital. It can serve as a reference point for future scientists, prior to performing an assay for assessing nano-titania’s genotoxic and cytotoxic effects, with the aim to obtain reliable answers.

Altogether, the uncontrolled parameters in assays prompt the need for a clear, invariable definition of titania’s toxicity. Implementing a proper methodology for toxicity assays of light-sensitive materials is of the utmost importance. A consensus on hazard assessments of nano-TiO_2_ must be reached after extensive scientific collaborations. Unfortunately, what the accuracy percentages reveal is that scientific competition and bias may take precedence over communal efforts to design adequate, errorproof protocols for toxicity assays of light-sensitive metal oxides, such as TiO_2_.

## 7. Conclusions and Outlook

This review has highlighted the impact, both positive and detrimental, of titanium dioxide’s structure and catalytic properties in the pharmaceutical, dermato-cosmetic, environmental safety, and food industries. Each sector can be improved through extensive research efforts.

In view of medicinal breakthroughs, it becomes clear that a balance is required between using nano-titanium dioxide as an anticancer agent and its cytotoxic potential on healthy cells. As a decontaminant, the metal oxide is highly efficient, but special care is needed when dispersing the catalytic slurries into the environment, to ensure the proliferation of ecosystems is not impacted long-term by the residual accumulation of this mineral. In the food packaging sector, AFP is structurally versatile and displays great potential to combat white pollution and food waste. In the food processing sector, however, the controversial use of E171 has reached a point of international disputes. As such, the authorities in charge of elucidating its mutagenic potential must also be responsible for raising societal awareness of its incorporation into alimentation.

Another challenge is posed by achieving control over the oxide’s reactivity. Given its multitude of uses linked to human exposure routes, it becomes vital to limit its catalytic activity, but preserve its light-scattering capacity. Scientific advances have been made towards coating titanium dioxide nanoparticles with inert shells. Experimental parameters are still in need of improvement, to accomplish more than a delay in titania reactivity, and possibly target an irreversible, complete catalytic inhibition.

At present, it is also difficult to assess what titania toxicity entails. Its dual behaviour towards light can mask the real experimental outcomes of toxicity assays that are based on the optical detection of cell viability parameters. This fact is concerning because many toxicity assays require protocolar adjustments for light-sensitive catalysts. In this respect, modern meta-analytical tools provide useful insights, by screening databases of experimental reports. Such analyses could guide future scientists on what experimental conditions need special calibrations for titanium dioxide and how to achieve them in the laboratory. The standpoint of the author is that joint scientific efforts are required to ascertain the mechanisms of titanium dioxide toxicity, as this influences all the abovementioned uses and breakthroughs.

## Data Availability

Not applicable.

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
