# Peer review of "Titanium Dioxide: Structure, Impact, and Toxicity"

_ijerph, 2022, doi:10.3390/ijerph19095681_

Round 1
Reviewer 1 Report
This manuscript presents a review on the latest findings on titania usage in industry and its possible impact and toxicity on human’s health and on the ecosystem. Because titania is a very popular material, thousands of reviews are published per year (5183 in 2021 and already 2023 in 2022, according to Scopus). Thus, it is hard to be able to give a significant and innovative contribution in this field. Beside this, this manuscript is well written and quite exhaustive, so in my opinion it is still worth publishing on Int. J. Environ. Res. Public Health. My main concern is about the real scientific knowledge of the author about the light-matter interaction phenomena and on the photocatalytic mechanism. The authors always refer to “electron transport”, while “excitation” or “transition” are more appropriate. The photocatalytic activity of titania is depicted like a “Holy Grail” able to react with any species absorbed on the surface. I suggest the author to read for example the review of Fujishima et al. / Surface Science Reports 63 (2008) 515–582, section 4- Fundamentals of photocatalysis, to improve this part, especially Page 2 lines 57-59. Please rewrite this sentence in a more scientific acceptable manner.
Other revisions:
Caption of Fig.3. I don’t understand why fig.3 should have to do with the crystallinity of TiO2. What I actually see is the possible effect of dopants on the bandgap electronic structure, i.e. formation of intra-band gap donor or acceptor levels. Please correct accordantly.
capture fig.1. please add TiO2 powder
Page 3 line 54: the band gap energy depends from the crystal structure and the particle size (see for example “Photochem. Photobiol. Sci., 10 (2011) 355). Typical values are 3.2 eV for anatase and 3.0 eV for rutile. Please correct and add reference.
Move caption of Fig.4 after the figure
Author Response
Response 1: I have updated the information regarding the light-matter interaction phenomena, detailing the photoexcitation phenomenon at titanium dioxide more accurately.
Response 2: The caption figures have also been detailed as per reviewer's instructions.
Response 3 : I have updated reference 11, in order to provide adequate references and band gap values for rutile and anatase (see chapter 1.2).
Reviewer 2 Report
The manuscript entitled: Titanium dioxide: structure, impact, and toxicity provides overall information on the structures, decontamination features, applications in sunscreen and food preservation, and toxicity of titanium dioxide. But the following major concerns should be addressed before its acceptance for publication:
- There are a lot of punctual and grammar mistakes in the manuscript. The author should make changes accordingly.
- Sentences should be formulated more clearly. For example - The ions expand in three dimensions 27 in an orderly but different fashion for each polymorph, which leads to distinct patterns of 28 crystallinity. What does different fashion mean?
- Sentences should be well-structured. For example, recent breakthroughs are herein summarized, with a focus on how restructuring the surface properties 1either enhances, or inhibits titania’s reactivity, depending on the role required. This particular sentence can be written better - Recent breakthroughs are summarised herein, focusing on whether restructuring the surface properties either enhances or inhibits titania’s reactivity, depending on the required role.
- In line 73, the author described that “Valence band holes, with their strong oxidant character, interact with defective cells, causing significant intracellular damage, to finally induce their necrosis”. However, the production of reactive oxygen species is responsible for cellular damage and necrosis. The author should replace and rewrite the sentences accordingly.
- In Line 112, it was mentioned that UV light decomposes sulfur mustard blistering agent and few citations were given. However, reference 32 is not related to the mentioned statement. Citation number 32 should be removed since the author there uses the sorption technique to remove the sulfur mustard. Similarly, in line 111, the also author mentioned photocatalysis technique is used to remove the toxins and cited reference 32. Therefore, reference 32 also should be removed. And, also reference no 33 and 35 is also not related to photocatalysis/UV light decomposition, these two references mention general aspects and analysis of degradation products of chemical warfare agents. Since the references are incoherent I strongly recommend the author to carefully check the references are relevant to the statement given.
- The author describes the use of photocatalytic properties of TiO2 in chemical warfare and water decontamination in the manuscript. I recommend the author add a separate sub-heading about microbial inactivation applications of TiO2. , which will be relevant to the application described.
- Regarding, the toxicity of TiO2 nanomaterials the author describes the use of different assays and the controversy regarding the assay. However, there are other good assays such as WST-1 assay and glutathione assay which could also monitor the cytotoxicity of the nanomaterials. For example, in this article Karthikeyan Thirunavukkarasu, J. Bacova, O. Monfort, E. Dworniczek, E. Paluch, M. Bilal Hanif, S. Rauf, M. Motlochova, J. Capek, K. Hensel, G. Plesch, G. Chodaczek, T. Rousar and M. Motola, Applied Surface Science, 2022, 579, 152145. The authors used WST-1 and glutathione assay to evaluate the cytotoxicity of different TiO2 nanomaterials in A549 cells. I recommend the author to add about these two types of assays too in the review.
- Also, the author mentioned the TiO2 is toxic due to the photocatalytic properties of TiO2 and sometimes TiO2 are toxic especially at higher concentrations of TiO2. It would be better if author distinguishes in separate sub-heading regarding toxicity such as toxicity of TiO2 without exposure to light and with exposure to light. Moreover, at lower concentration TiO2 is not so toxic, so author could distinguish in separately about dose dependent toxicity.
Author Response
Response 1 : Throughout the text you will see amendments to any punctual or grammar mistakes.
Response 2 : Sentences have been reformulated at times to make them more clear.
Response 3 : I have restructured that particular sentence in the appendix and also restructured sentences throughout the manuscript to provide more clarity.
Response 4 : The sentence has been reformulated to include the presence of reactive oxygen species and their effect on cellular necrosis.
Response 5 : To improve the relevance of the references that relate to titanium dioxide as a decontaminant for chemical and biological warfare agents, references 32-35 have been updated and the information has been described in the text.
Response 6 : The capacity of titanium dioxide to inactivate biological warfare agents has been added in section 2.1. The new articles referenced mention the metal's capacity to decontaminate both biological and chemical warfare agents.
Response 7 : The WST-1 assay and glutathione assay have been added in the manuscript in Table 2, alongside a short explanation on their detection target.
Response 8 : I have distinguished as best as possible between dose dependent toxicities. In chapter 5, additional subsections have been titled 5.2.1. Concentration discrepancies and 5.2.2. Human toxicity models in different concentrations.The human model of intestinal cells that I have cited in section 5.2.2 focuses more on the barrier to titanium dioxide’s reactivity rather than its possibility to accumulate on this type of cells. Therefore, the initial structure by which toxicity is depicted in terms of route of intake – inhalation, ingestion, dermal - was slightly amended.